

# Co-expression analysis and ceRNA network reveal eight novel potential lncRNA biomarkers in hepatocellular carcinoma

Ren-chao Zou[1,*], Zhi-tian Shi[1,*], Shu-feng Xiao[1,2], Yang Ke[1], Hao-ran Tang[1], Tian-gen Wu[1], Zhi-tang Guo[1], Fan Ni[1], Sanqi An[3] and Lin Wang[1]

[1] Department of Hepatobiliary Surgery, the Second Affiliated Hospital of Kunming Medical University, kunming, China
[2] Department of General Surgery, Puer People's Hospital, Puer, China
[3] Center for Stem Cell Biology and Tissue Engineering, Key Laboratory of Ministry of Education, Sun Yat-Sen University, Guangzhou, China
[*] These authors contributed equally to this work.

## ABSTRACT

**Background**. Hepatocellular carcinoma (HCC) is the most common primary liver cancer in the world, with a high degree of malignancy and recurrence. The influence of the ceRNA network in tumor on the biological function of liver cancer is very important, It has been reported that many lncRNA play a key role in liver cancer development. In our study, integrated data analysis revealed potential eight novel lncRNA biomarkers in hepatocellular carcinoma.

**Methods**. Transcriptome data and clinical data were downloaded from the The Cancer Genome Atlas (TCGA) data portal. Weighted gene co-expression network analysis was performed to identify the expression pattern of genes in liver cancer. Then, the ceRNA network was constructed using transcriptome data.

**Results**. The integrated analysis of miRNA and RNAseq in the database show eight novel lncRNAs that may be involved in important biological pathways, including TNM and disease development in liver cancer. We performed function enrichment analysis of mRNAs affected by these lncRNAs.

**Conclusions**. By identifying the ceRNA network and the lncRNAs that affect liver cancer, we showed that eight novel lncRNAs play an important role in the development and progress of liver cancer.

Corresponding authors
Sanqi An, ansq@mail2.sysu.edu.cn, ansanqi2016@gmail.com
Lin Wang, wanglinfey@126.com

## INTRODUCTION

Liver cancer, which is most common among male patients, is the high leading cause of death in men worldwide (*Wang et al., 2015*). In the United States, there are as many as 42,030 new cases and 31,780 deaths related to liver cancer every year, based on the latest statistics record (*Siegel, Miller & Jemal, 2019*). The degree of malignancy of cancer can be determined by study of the histology of the tumor, and patients can be divided into three classes, low, medium, and high according to the degree of malignancy. Clinically, primary

liver carcinoma is considered to be one of the most common malignant tumors, and about 90% of these tumors are hepatocellular carcinoma (HCC). In cases of patients diagnosed with HCC, 50.7% of them achieve a 5-year survival rate, while the median survival time is 60 months (*Lee et al., 2006*). The prognosis of HCC patients is related to the patient's disease stage. Currently, the tumor-node-metastasis (TNM) pathological staging standard developed by the American Joint Committee on Cancer (AJCC) is the most commonly used malignant tumor staging system worldwide. However, the relationship between long non-coding RNAs (lncRNAs) and tumor staging has raised concern among researchers (*Chen et al., 2015*; *Ou et al., 2018*; *Yao et al., 2018*), suggesting that many lncRNAs play progressive key roles in HCC development (*Chen et al., 2016a*). For example, the MALAT-1 gene is upregulated in HCC and also correlates with prognostics and recurrence (*Guerrieri, 2015*; *Lai et al., 2012*); the overexpression of the HULC gene may reduce the mir-372 gene, while at the same time may promote reprogramming during tumorigenesis (*Du et al., 2012*); and DANCR induces stemness features and could serve as a potential prognostic marker and therapeutic target for HCC (*Yuan et al., 2016*). Previous studies have mainly focused on the single biomarker use of miRNAs in HCC.

The weighted gene co-expression network analysis (WGCNA) is a popular bioinformatic method used in the construction of gene networks and the detection of gene modules and their phenotypic traits (*Langfelder & Horvath, 2008*; *Yin et al., 2018*). In this study, we identified eight novel lncRNAs correlated with TNM using WGCNA. Additionally, functional enrichment analysis shows that these eight novel lncRNAs play an important role in the regulation of gene expression that affect development and progression of liver cancer.

## MATERIALS AND METHODS

### Data preprocessing and differential gene selection

The clinical data, RNAseq data and microRNA data of LIHC were downloaded from the The Cancer Genome Atlas (TCGA) database (*Akbani et al., 2014*) (Table S1). There were 49 pairs of microRNA samples and 50 pairs of RNAseq samples in total. The variation between the RNA and microRNAs data was calculated using EdgeR package (*Reimers & Carey, 2006*; *Varet et al., 2016*). Only microRNAs recorded with first 10 and last 10 FC values were selected for subsequent analysis (*Li et al., 2018*; *Shao & Li, 2019*). Because these biggest expression changed miRNA may have real major function in HCC. For RNAseq data, the EdgeR package was used to calculate the differential correlationship, and the threshold value for FDR was set at <0.01, | logFC |>1 (Table S2). Clinical data were used to calculate the correlation matrix of clinical information for integrated analysis. We used the "heatmap.2" function in the "gplots" package to create the heatmap.

### Determination of ceRNA

Using microRNAs and differential expression genes as input data for target prediction, the RNA22 program was used to predict binding sites of microRNAs based on their sequence characteristics (*Loher & Rigoutsos, 2012*). Based on these ceRNA interactions, we obtained all the mRNA-miRNA pairs with sharing the number of common miRNAs. Because the

number of the ceRNA pairs is obey hypergeometric distribution. We estimated their statistical significance by a hypergeometric test. The potential top 20 different expression miRNA with hypergeometric test in ceRNA network *P* value less than 0.05 was obtained (*Li et al., 2018*; *Shao & Li, 2019*). Specific formulas, such as the differential expression matrix of lncRNA, were used to get the Pearson correlation coefficient (PCC) (Table S3). Based on ceRNA's mechanism, the expression matrix (EM) for lncRNA to bind between themselves resulted in a PCC of EM >0. Co-expression is one of the features of ceRNA network on account of their interactions. The final ceRNA pair was obtained by intersecting the ceRNA threshold with cutoff *p*-value <0.05. Cytoscape v3.0 was used to construct the ceRNA network. The overall Kaplan–Meier (KM) survival analysis in each subtype was performed using GEPIA2 database (Tang, 2017).

## WGCNA model related computing

Weighted gene co-expression network analysis (WGCNA), a bioinformatic method used to find correlation patterns among genes, was utilized in this research. WGCNA assumes that gene expression networks is scale-free, it uses a 'soft' threshold to determine the weights of the edges connecting genes and merge individual genes to a module. An appropriate soft threshold will make the co-expression network closer to a scale-free network. Then we constructed a signed weighted co-expression network using WGCNA based on the gene expression value across the TCGA samples. We obtained 60 co-expression modules according to the correlations of fpkm value among samples. Each module is represented by an *value* belongs to the '*eigengene*'. This *value* is identified from the *principal component analysis* (PCA) of all the gene expression value in the module. Then, we find the relationship between modules and the trait, Eventually, unsupervised WGCNA identified major lncRNAs expression modules with different degrees of correlation to TNM staging using WGCNA R software (*Langfelder & Horvath, 2008*).

## Prediction of interrelationship between lncRNA-related mRNAs

Using RNAseq data from TCGA, the correlation prediction of the lncRNA-lncRNA network and lncRNA-mRNA network was constructed. The cutoff of PCC was 0.7. We retained the intersection with the lncRNA using a cutoff of the top 20% degree. Ultimately, we found eight potentially important lncRNAs with high degrees of connection in the ceRNA network and high correlation at the expression level.

## Gene ontology and KEGG enrichment analysis

Gene ontology (GO) annotation analysis was performed using DAVID software (*Huang da, Sherman & Lempicki, 2009a*). Gene functions for these important indirectly regulated mRNA genes elucidate that this mRNA regulated by lncRNA may have some key biological functions (*Huang da, Sherman & Lempicki, 2009a*; *Huang da, Sherman & Lempicki, 2009b*). String database was used for protein-protein analysis (*Szklarczyk et al., 2017*).

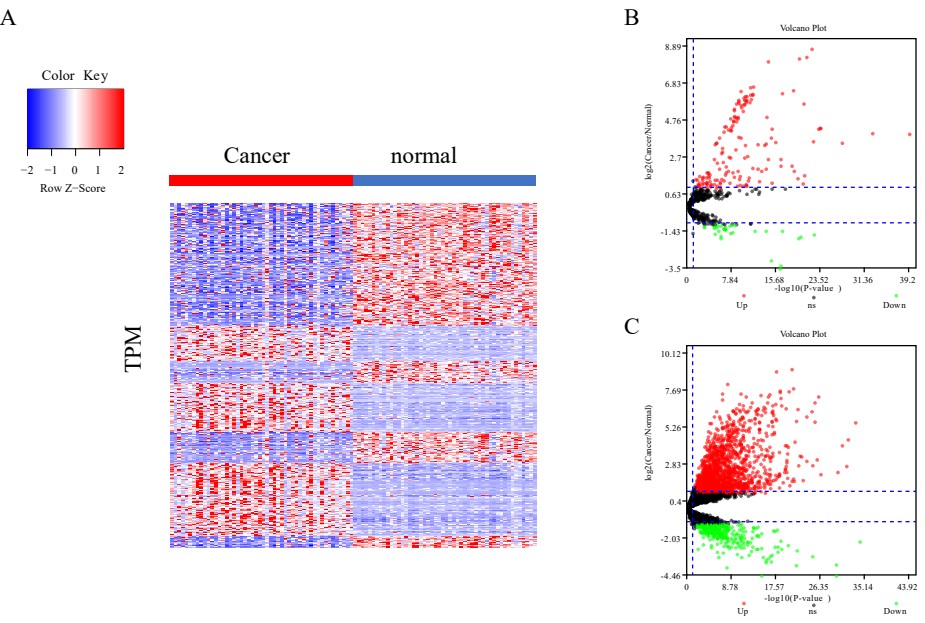

**Figure 1** **Different expression of mRNA, miRNA and lncRNA levels between tumor and normal samples.** (A) Heatmap showing 2,763 different expression of mRNA between tumor and normal samples in LIHC. (B) Volcano map for 310 different expression miRNA. Dots in red and green indicate high and low expression of miRNA in cancer, respectively. (C) Volcano map for 1,962 different LncRNA. Dots in red and green indicate high and low expression of lncRNA in cancer, respectively.

## RESULTS

### Analysis of differential miRNAs and differential lncRNAs

The differential expression of miRNAs and lncRNAs between normal samples and cancer samples was calculated separately using the EdgeR package (*Chen et al., 2017*; *Law et al., 2016*; *Maza, 2016*). There were 1962 significant differential expression genes in lncRNAs, and 310 differentially expressed miRNA genes found between healthy and cancer-treated samples. Almost half of the mRNAs are upregulated and half downregulated (Fig. 1A). Most of the gene expression for lncRNAs and miRNAs was upregulated in cancer-treated samples (Figs. 1B–1C).

### Weighted gene co-expression network analysis of lncRNA

To determine if any of the identified coexpression modules were associated with TNM stage, we calculated the PCC between the MEs and TNM stage. All lncRNAs were merged to 60 modules according the degree of coexpression across the data set in WGCNA. As in the previous study, we assigned each coexpression module an arbitrary color for reference (*Di et al., 2019*; *Zhussupbekova et al., 2016*). The hierarchical clustering dendrogram of the eigengenes shows the module size (the number of genes per module) and relationships among these modules (Fig. 2A). Most modules had minimal relationships with each other.

Comparison of module-characteristic eigengenes showed TNM was best correlated with the module MElightpink4 ($p = 4E–04$) and MElightcyan ($P < 0.006$), composed of 125
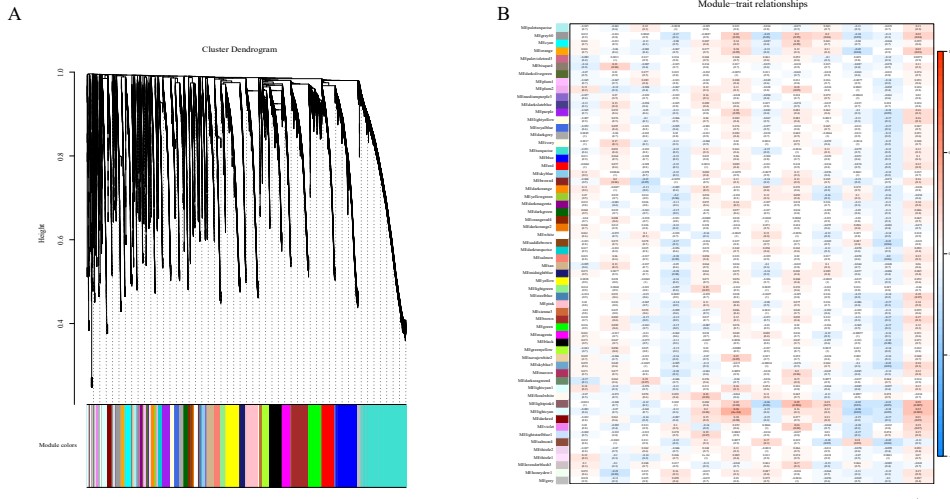

**Figure 2** **Results of Weight Gene Co-expression Network Analysis (WGCNA).** (A) shows the clustering dendrogram of co-expression lncRNA based on topological overlap. (B) Module-TNM stage correlative analysis. Each row corresponds to a module eigengene, each column corresponds to a TNM stage. Heatmap block with *p*-values and correlation coefficient. The red box in the figure shows the module with higher correlation coefficient in the three stages of TNM. The blue box in the figure shows the module with negative correlation coefficient in the three stages of TNM.

lncRNAs (Fig. 2B). The PCC values ranged from −1 to +1 depending on the power of the relationship. A positive value indicated that the lncRNA within a particular co-expression module increased as the TNM increased, whereas the opposite occurred if the sign of the PCC was negative. We learned that the correlationship between the module and TNM stage with PCC value was accompanied by the corresponding *P*-value in brackets. These modules included genes that were co-expressed in a particular TNM stage can be used to represent the TNM stage of HCC development (Fig. 2B). These gene may be the risk factors and therapeutic targets in the treatment of HCC.

## ceRNA network of lncRNA reveals potential biomarkers in liver cancer

As shown in Fig. 3A, ceRNA network was constructed by high degree lncRNAs. The topological characteristics of ceRNA network were analyzed with degree, betweenness, and closeness (Table S4). Many genes in ceRNA network are with high degree, betweenness, or closeness like AC016773.1, AC145285.2, LINC01569 and DANCR, and so on. This implied ceRNA network may regulate many gene expression through these lncRNAs to have effect on progression of liver cancer. To verify the function of these lncRNAs, Kaplan–Meier survival analysis was perform ed for expression level of these lncRNAs. The results of the survival analysis presented in Figs. 3B and 3C show patients with high expression level of DANCR or AL671710.1 have poor prognosis. It is a validation of our analysis and two of the lncRNAs may affect prognosis.

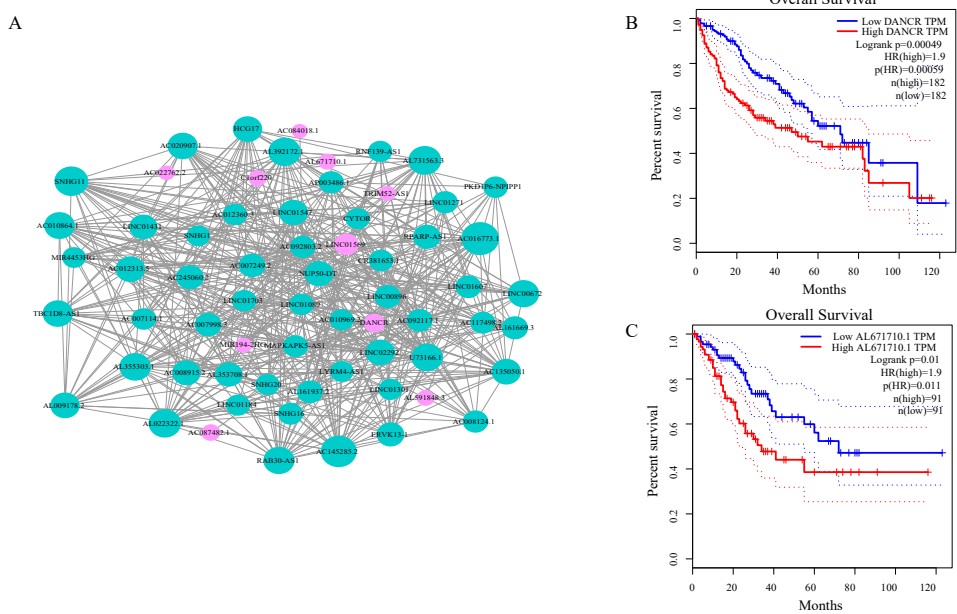

**Figure 3** **CeRNA network in LIHC and survival curves.** (A) Red dots represent overlaping lncRNA selected from co-expression analysis and RNA22 which can identify microRNA binding sites. The bigger the dots, the higher the degree value, the more important nodes in the network. (B) Survival curve of patients with high expression of DANCR and low expression of DANCR. Survival curve of patients with high expression of AL671710.1 and low expression of AL671710.1.

## The interactions between lncRNAs and mRNAs

Combining WGCNA with the results of ceRNA prediction analysis, lncRNA was combined with correlation prediction to find eight important lncRNAs including AL671710.1, TRIM52-AS1, C1orf220, AC022762.2, DANCR, LINC01569, AC084018.1 and MIR194-2HG. Among these lncRNAs, downregulation of TRIM52-AS1 play key role in renal cell carcinoma (*Liu et al., 2016*). DANCR increases stemness features of hepatocellular carcinoma (*Yuan et al., 2016*). DANCR is also associated with various cancer (*Lu et al., 2018a*; *Lu et al., 2018b*; *Mao et al., 2017*; *Sha et al., 2017*; *Xu et al., 2018*; *Yuan et al., 2016*). However, there is little known about the function of other six lncRNAs in cancer. There were 124 genes targeted by these eight lncRNAs (Figs. 4A–4H), including EIF3 which functions during the initiation phase of translation. TRIM52-AS1 may influence cancer behavior and function through interactions with regulator EIF3. EIF3 plays a key role in human diseases (*Gomes-Duarte et al., 2018*; *Valasek et al., 2017*). Also, there were 30 genes targeted by lncRNA ac084018.1, including m6A reader methyltransferase like 3 (METTL3). As reported in previous research, METTL3 promotes liver cancer progression through YTHDF2 (*Balacco & Soller, 2019*; *Berlivet et al., 2019*; *Chen et al., 2018*; *Weng et al., 2018*). DANCR and AL671710.1 also have crucial roles through certain key genes (Figs. 4C, 4E).

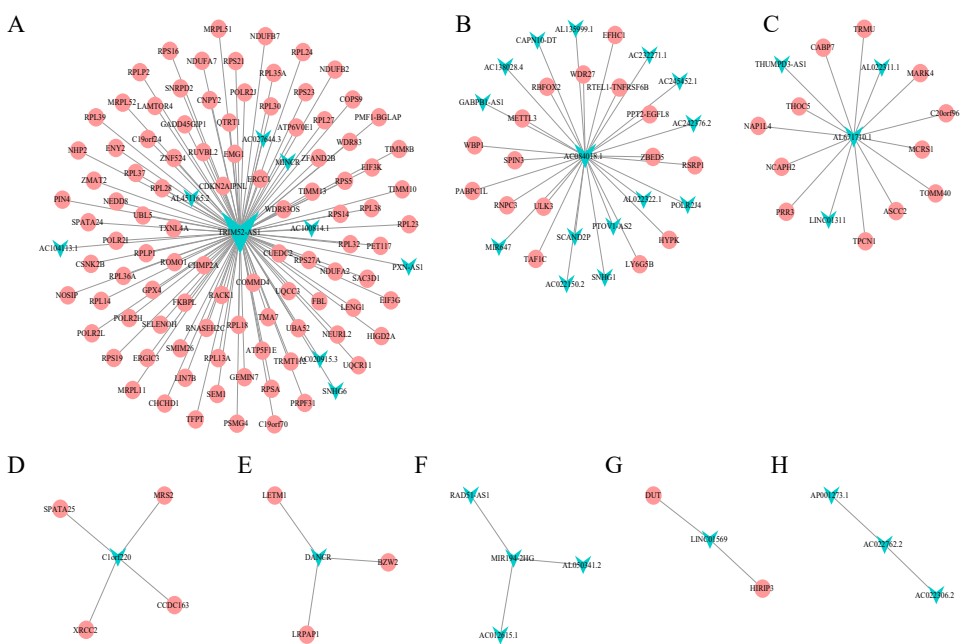

**Figure 4 Interaction of eight lncRNA and target mRNA.** The green arrows represent the lncRNA, and the pale red dots represent the mRNA. It can be seen that the eight important lncRNA AL6717.1, TRIM52-AS1, C1220C1, DANCR, LINC01569, AC084018.1 participated in many regulation of mRNA.

## GO and KEGG pathway enrichment analyses of lncRNA-targeted gene

Additionally, we performed GO and KEGG enrichment analysis of the mRNAs in the network (Figs. 5A–5C). We analysed target genes of the lncRNA based on their enrichment scores for associated GO terms and KEGG pathways using David tools (*Huang da, Sherman & Lempicki, 2009a*; *Huang da, Sherman & Lempicki, 2009b*) . The GO and KEGG enrichment analysis concerning the target genes of lncRNAs indicated that the top regulated pathways of lncRNAs were Huntington's disease, RNA polymerase and Pyrimidine metabolism, and the top regulated functions of lncRNAs were SRP-dependent co-translational protein targeting to membrane, translational initiation, viral transcription, nuclear-transcribed mRNA catabolic process, and nonsense-mediated decay. Furthermore, those essential mRNA may interact with each other and function in HCC (Fig. 6A). We can conclude that those lncRNAs affect HCC through the functions and pathways listed above (the flow chart was depicted in Fig. 6B).

## DISCUSSION

Hepatocellular carcinoma (HCC) is one of the primary causes of cancer-related death worldwide (*Balogh et al., 2016*). Many genes influence HCC, TP53 tumor-suppressor gene, p16INK4A and Rb-associated with various risk factors have been largely reported (*Bae et al., 2016*; *Buendia, 2000*; *Nishida & Fukuda, 2001*; *Peng et al., 2013*; *Wang et al., 2017*; *Zhang et al., 2014*). Some even take part in the cancer biography progress through lncRNA (*Su et al., 2017*). Long non-coding RNAs (lncRNAs) which are transcribed but do not

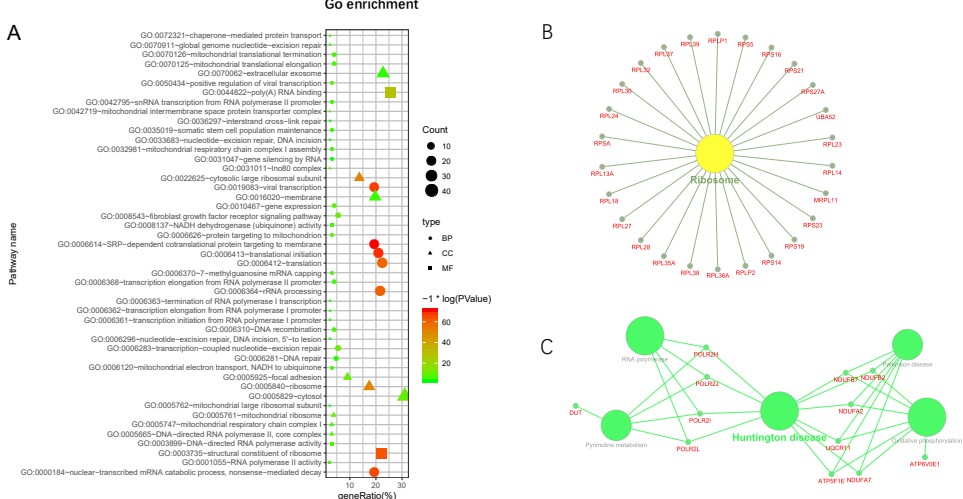

**Figure 5 GO and KEGG pathway enrichment analysis of target gene.** (A) GO enrichment of mRNA interact with lncRNA. Points of different shapes represent BP, CC and MF of GO, and the size of points represents the number of gene enriched in the GO function; the number of different colors represents P value. (B) The KEGG pathway is enriched in ribosome and in Huntington's disease.

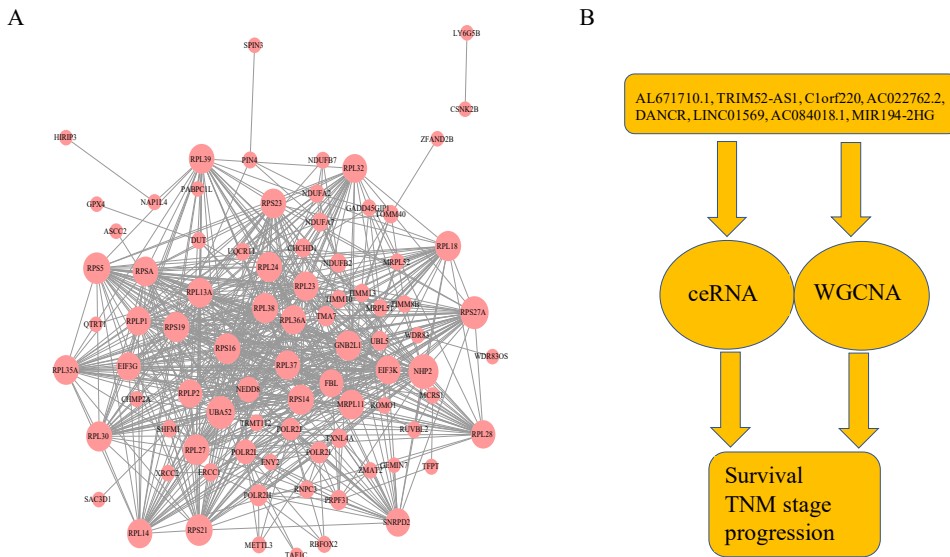

**Figure 6 The protein–protein interactions of RNA and conclusion summary.** (A) The PPI network of RNA with degree bigger than 10 in string database. The larger the dots in the graph, the more interaction between the RNA corresponding proteins and other proteins. (B) Conclusion summary.

encode proteins, play key roles in HCC development (*Abbastabar et al., 2018*). These include MALAT-1 and also NEAT-2, which regulates splicing factors mostly situated in nuclear speckles. In addition, MALAT-1 is a biomarker in various cancers including HCC (*Lai et al., 2012*; *Wang et al., 2016*). LncRNA GAS5 is a biomarker and have potential

applications in HCC therapy (*Fang et al., 2019*). Using WGCNA and hypergeometric test analysis, we found eight lncRNAs with important functions: AL671710.1, TRIM52-AS1, C1orf220, AC022762.2, DANCR, LINC01569, AC084018.1 and MIR194-2HG. TRIM52-AS1 is one of the eight lncRNAs that has been reported as a function of a tumor suppressor (*Liu et al., 2016*; *Zhang et al., 2017*).Targeted by MYC, DANCR promotes cancer (*Chen et al., 2016b*; *Dhanasekaran et al., 2017*; *Huang, Deng & Zhou, 2013*; *Kron et al., 2012*; *Lu et al., 2018b*). Taken together, we concluded that these lncRNAs may function as a potential tumor regulator in HCC.

Additionally, some lncRNAs were associated with the TNM stage in HCC tissues (*Abbastabar et al., 2018*). The American Joint Committee on Cancer (AJCC) stratifies patients using a Tumor-Node-Metastasis (TNM) classification, representing a group of models useful in the assessment of tumor extension (*Selcuk, 2017*; *Tellapuri et al., 2018*). Among several staging systems, the TNM system was one of the most widely accepted, and had a higher prognostic competency than the other systems (Prognosis Evaluation in Patients with Hepatocellular Carcinoma after Hepatectomy, Comparison of BCLC and Hangzhou Criteria Staging Systems). In our work, we found two lncRNAs modules associated with the TNM stage. Those lncRNAs may function as the biomarker of node size and metastasis status in HCC. Systematic analysis of transcriptomics data reveal those novel potential therapeutic target may be involved in cancer-related pathway in liver cancer. Our study has limitations, the specific mechanism of these lncRNAs remains unexplored.

## CONCLUSIONS

In summary, our results demonstrated that lncRNAs AL671710.1, TRIM52-AS1, C1orf220, AC022762.2, DANCR, LINC01569, AC084018.1, and MIR194-2HG play an essential role in the HCC stage, and their targeted mRNA have key functions in HCC. Those lncRNAs might be a novel prognostic biomarker for HCC.

### Funding

This work was supported by National Natural Science Foundation of China (No. 81660399 and 81860423), Yunnan Provincial Clinical Center of Hepato-biliary-pancreatic Diseases [no specific number], the Doctor Newcomer Award of Yunnan Province in 2017 [no specific number], and Ph.D. Student Innovation Fund of Kunming Medical University (No. 2019D004). The funders had no role in study design, data collection and analysis, decision to publish, or preparation of the manuscript.

### Grant Disclosures

The following grant information was disclosed by the authors:
National Natural Science Foundation of China: 81660399, 81860423.
Yunnan Provincial Clinical Center of Hepato-biliary-pancreatic Diseases.
Doctor Newcomer Award of Yunnan Province in 2017.
Ph.D. Student Innovation Fund of Kunming Medical University: No. 2019D004.

## Competing Interests

The authors declare there are no competing interests.

## Author Contributions

- Ren-chao Zou performed the experiments, analyzed the data, contributed reagents/materials/analysis tools, prepared figures and/or tables, authored or reviewed drafts of the paper, approved the final draft.
- Zhi-tian Shi performed the experiments, analyzed the data, prepared figures and/or tables, approved the final draft.
- Shu-feng Xiao analyzed the data, contributed reagents/materials/analysis tools, prepared figures and/or tables, approved the final draft.
- Yang Ke analyzed the data, prepared figures and/or tables, approved the final draft.
- Hao-ran Tang analyzed the data, prepared figures and/or tables, authored or reviewed drafts of the paper, approved the final draft.
- Tian-gen Wu analyzed the data, prepared figures and/or tables, approved the final draft, review and editing.
- Zhi-tang Guo prepared figures and/or tables, approved the final draft, validation.
- Fan Ni analyzed the data, prepared figures and/or tables, approved the final draft, data curation.
- Sanqi An conceived and designed the experiments, authored or reviewed drafts of the paper, approved the final draft.
- Lin Wang conceived and designed the experiments, prepared figures and/or tables, authored or reviewed drafts of the paper, approved the final draft.

## Data Availability

The raw measurements are available in the Supplementary Files.

## Supplemental Information

Supplemental information for this article can be found online at http://dx.doi.org/10.7717/peerj.8101#supplemental-information.

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
