# Peer review of "Co-expression analysis and ceRNA network reveal eight novel potential lncRNA biomarkers in hepatocellular carcinoma"

_PeerJ, doi:10.7717/peerj.8101_

## Round 0.1 · original submission · Major Revisions

In this work, you performed a valid analysis of ceRNAs in HCC. identifying eight lncRNAs implicated in HCC development. The results are very interesting for the scientific community and for the journal.
The manuscript is generally well written.

However, I agree with comments of reviewers about the need of revision of the actual form of the manuscript. In particular, I suggest you to label properly the panel of any figure to make clearer the results.

As example, in figure 1 I would suggest to specify "Normal" and "Cancer" on the two plots and indicate the meaning of any color.

I also suggest you to discuss potential directions for validation of the identified targets in future studies.

Reviewer 1 ·

Basic reporting

The manuscript consists of lots of grammatical errors which hinders the comprehension. The authors should address this.

The authors have provided a comprehensive background of the subject with sufficient literature references.

The authors need to improve the presentation structure of figures and tables. In Figure1 the authors should mentioned what each color in the volcano plot represent.

Experimental design

The authors should mention the rationale for selecting only the microRNAs with first 10 and last 10 FC values for subsequent analysis.

Validity of the findings

The authors should discuss about the impact and novelty of their study, including the limitations of the study.

Reviewer 2 ·

Basic reporting

The language is easy to follow, but the background of HCC and WGCNA is no sufficient.
The article structure, figures and table layout is good.

Experimental design

The experimental design lack the validation experiment.

Validity of the findings

No comment.

Additional comments

This article aim to reveal potential novel lncRNA biomarkers in HCC. The whole layout of manuscript is reasonable. But the manuscript lack the verification of key targets.

·

Basic reporting

The study is clear, and can do some good impact to the current knowledge.
But we need to consider adding and correcting few things to make it ready for publication in case you consider it and agree with my simple opinion.

-Language:The language is acceptable and well written. But, would recommend few more revisions and correct few minor things.

-The Title:(Co-expression analysis and ceRNA network reveals potential eight novel lncRNA biomarkers in hepatocellular carcinoma) Please remove the (S) from the verb(reveals)to make it Grammatically correct.

-introduction: Hepatocellular cancer [HCC] is the most common primary liver cancer in the world, with high degree of malignancy? they need to define the (degree of malignancy?) and embed the answer in your introduction text.

-In the introduction/ Lines: 2-3: (is the second highest leading cause of death after cardiovascular disease) please correct this part using the following link CDC reports:
https://www.cdc.gov/nchs/data/nvsr/nvsr67/nvsr67_06.pdf


-In the introduction fifth and sixth lines/L5-6:
[primary hepatocellular carcinoma (PLC) is considered to be one of the most common malignant tumors, and about 90% of these tumors are hepatocellular carcinoma (HCC)]!
Please consider making some correction to the expression way of the whole sentence and the (abbreviation PLC) and you might consider using the term (primary hepatic cancer) or (primary liver cancer). Because the HCC is the subdivision of that.


-In the introduction: Lines 22 and 23:
(Additionally, functional enrichment analysis shows that these eight novel lncRNA play
23 an important role in the development and progression of liver cancer.)
Please refer to the parts of your study supporting this fact so that would be stronger introduction.


-Materials and Methods: Please provide more details about using WGCNA and hypergeometric test analysis?


-Data preprocessing and differential gene selection:
Lines(26-27) (The clinical data, RNAseq data and microRNA data of LIHC were downloaded from the TCGA database): Please consider Spelling out acronyms the first time you use them, even if they are commonly used ones, for example: The Cancer Genome Atlas(TCGA).

-The (Volcano plots) of differentially expressed RNAs already used which is a very nice work, great add to the manuscript.

-You have to add more details to the (Heatmap plots) of differentially expressed mRNAs.
/ More details about the axis representing the samples?

-Add more details explaining and simplifying your tables and figures in the captions please. Especially Figure 1. (Different expression of miRNA and lncRNA).

-They should consider adding small paragraph summarizing the conclusion, although it’s already included in different parts of the results, but it would be a great add to the manuscript.

Experimental design

-Materials and Methods: details about how they used(WGCNA and hypergeometric) test analysis?

-We need more details to the (Heatmap plots) of differentially expressed mRNAs.
/ More details about the axis representing the samples.

-More details explaining and simplifying your tables and figures in the captions please. Especially Figure 1. (Different expression of miRNA and lncRNA).

-Summarizing the conclusion by the end would be a good add.

Validity of the findings

Please consider doing some statistical review by a specialist and reviewing the validity of the graphs.

Additional comments

Dear authors,

I reviewed your work (Co-expression analysis and ceRNA network reveals potential
eight novel lncRNA biomarkers in hepatocellular carcinoma) It was interesting, but there were some problems to be solved. I want to review the revised paper again.

Minor points. We need requiring further details and corrections:

-The Title:(Co-expression analysis and ceRNA network reveals potential eight novel lncRNA biomarkers in hepatocellular carcinoma) Please remove the ( S ) from the verb(reveals)to make it Grammatically correct.

-introduction:
Hepatocellular cancer [HCC] is the most common primary liver cancer in the world, with high degree of malignancy? Please define the (degree of malignancy?) and embed the answer in your introduction text.

-In the introduction/ Lines: 2-3: (is the second highest leading cause of death after cardiovascular disease) please correct this part using the following link CDC reports:
https://www.cdc.gov/nchs/data/nvsr/nvsr67/nvsr67_06.pdf


-In the introduction fifth and sixth lines/L5-6:
[primary hepatocellular carcinoma (PLC) is considered to be one of the most common malignant tumors, and about 90% of these tumors are hepatocellular carcinoma (HCC)]!
Please consider making some correction to the expression way of the whole sentence and the (abbreviation PLC) and you might consider using the term (primary hepatic cancer) or (primary liver cancer).


-In the introduction: Lines 22 and 23:
(Additionally, functional enrichment analysis shows that these eight novel lncRNA play
23 an important role in the development and progression of liver cancer.)
Please refer to the parts of your study supporting this fact so that would be stronger introduction.

-Materials and Methods: Please provide more details about using WGCNA and hypergeometric test analysis?

-Data preprocessing and differential gene selection:
Lines(26-27) (The clinical data, RNAseq data and microRNA data of LIHC were downloaded from the TCGA database): Please consider Spelling out acronyms the first time you use them, even if they are commonly used ones, for example: The Cancer Genome Atlas(TCGA).

-The (Volcano plots) of differentially expressed RNAs already used which is a very nice work in your paper.

-You have to add more details to the (Heatmap plots) of differentially expressed mRNAs.
/ More details about the axis representing the samples?

-Add more details explaining and simplifying your tables and figures in the captions please. Especially Figure 1. (Different expression of miRNA and lncRNA).

-Results: Are well stated.

-Please consider adding small paragraph summarizing the conclusion, although it’s already included in different parts of the results, but it would be a great add to your manuscript.

Reviewer 4 ·

Basic reporting

no comment

Experimental design

no comment

Validity of the findings

no comment

Additional comments

In this paper, Zou et al. performed integrated analysis of ceRNAs in Hepatocelluar carcinoma (HCC), and identified eight lncRNAs which were involved in important pathways. This is interesting work for research of HCC. However, some issues must be addressed as follows:
1. At the Abstract part, authors should revised to provide more description about Research methods, major Results, Significance, and Conclusion, not Introduction. The statements in the abstract that introduce HCC, ceRNA and lncRNA can be concatenated into one or two sentences.
2. The next sentence in the line 8 of Introduction should not be a transitional relationship with the previous sentence. “However” is inappropriate.
3. The sentence in lines 29 and 30 in 2.1 is somewhat confusing. How to get CeRNA? Method for cutoff 0.01?
4. The sentence beginning with “Specific” in line 39 of 2.2 needs to enumerate all the detailed formula descriptions used in the study, rather than special cases. What is co-expression effect?
In this 2.2, authors utilized RNA22 method to performed miRNA-target identification. Why not use miRcode、TargetScan or miRanda methods. Please explain it.
5. In 2.3, authors should describe the mechanism of action of WGCNA more clearly.
6. 2.5 The first two sentences are repeated and can be appropriately modified.
7. The meaning of “eigengenes” (Line 71) in 3.2? Figure 2 Since the meaning of the red box is explained, the meaning of the blue box should also be explained. And why the module analysis is performed in the Results? The conclusion of this part?
The relation between lncRNA modules and TNM stage must be explained clearly, and many sentences in this section belong to Methods part, not Results part.
8. Construction of a ceRNA network of lncRNA part belong to Methods Part, not Results part. The authors must revise this part and explain the biological meanings of the ceRNA network.
9. what is the relation and difference between networks from Figure 3 and Figure 4. And how to get eight lncRNA-cored networks in Figure4.
10. The description of lncRNA in the legend in Figure 4 does not match the figure (the triangle is in the figure, the arrow shape in the figure), and the description of lncRNA and mRNA should be the same shape or color.
11. Should the C diagram in Figure 5 be presented separately as Figure 6.

---

## Round 0.2 · Minor Revisions

The paper is improved but there are still some issues to address. In particular, please answers to the comments of reviewer 4 and re-submit.

Reviewer 2 ·

Basic reporting

This paper is clear and unambiguous, professional English used throughout.

Experimental design

The paper had a unequivocal aim and scope.

Validity of the findings

This study makes sense in the field of liver cancer research.

Additional comments

This study is very meaningful in the field of liver cancer research, it is recommended to accept.

Reviewer 4 ·

Basic reporting

More biological meanings of these eight lncRNAs were missing.

Experimental design

no comment

Validity of the findings

no comment

Additional comments

In this revision, the authors addressed some of my comments, however, most issues were not addressed or revised. So major revision need to be performed again before publication.

Q1. For the Abstract issue, the authors have re-written this part, including the background, methods, results and conclusions. I think the background part is too long than others.

Q3. For DE genes, the cutoff was defined as FDR<0.01 and |logFC|>1. However, for DE miRNAs, the totally different cutoff was defined. Why? Explain it, or discuss it.

Q4. What is the definition “co-expression effect”? The word “effect” is confusing, whether other articles used this word and provided the reference. More importantly, other miRNA-target databases must be discussed in your manuscript.

Q7. The section 3.2 was totally not revised. This part must be revised to provide more description about the Figure 2 or conclusions, and more References must be added to explain these results.

Q8. For one biological network, the basic attributes, such as degree and betweeness, must be displayed and analyzed. But this analysis was missing.

New comments. The description or explanations for the eight lncRNAs should added, and more references must be added to explain the relationship between these lncRNAs and hepatocellular carcinoma. Whether some lncRNAs have be identified as tumor bio-marker? And which lncRNA is novel bio-marker for tumor or hepatocellular carcinoma? The authors must display it.

---

## Round 0.3 · accepted · Accept

Thank you for addressing the second round of comments. The manuscript is improved and looks to me now suitable for publication.